# Incorporating Pragmatic Reasoning Communication into Emergent Language

**Yipeng Kang, Tonghan Wang**
Institute for Interdisciplinary Information Sciences
Tsinghua University
Beijing, China
{kyp13, wangth18}@mails.tsinghua.edu.cn

**Gerard de Melo**
Hasso Plattner Institute
University of Potsdam
Potsdam, Germany
gdm@demelo.org

## Abstract

Emergentism and pragmatics are two research fields that study the dynamics of linguistic communication along substantially different timescales and intelligence levels. From the perspective of multi-agent reinforcement learning, they correspond to stochastic games with reinforcement training and stage games with opponent awareness. Given that their combination has been explored in linguistics, we propose computational models that combine short-term mutual reasoning-based pragmatics with long-term language emergentism. We explore this for agent communication referential games as well as in Starcraft II, assessing the relative merits of different kinds of mutual reasoning pragmatics models both empirically and theoretically. Our results shed light on their importance for making inroads towards getting more natural, accurate, robust, fine-grained, and succinct utterances.

## 1 Introduction

In many linguistic theories (Zipf, 1935; Lewis, 1969; Grice, 1975), language is viewed as a special kind of social coordination system, in which multiple agents make interdependent decisions of how to express and comprehend messages in order to successfully communicate real-world information. Drawing on advances in artificial intelligence, recent work considers referential games (Lazaridou et al., 2018) to model language learning from raw sensory input, adopting techniques from computer vision, language processing, and multi-agent reinforcement learning. Considering agent communication issues in multi-agent learning not only benefits the coordination of agents for tasks with common objectives, but also manifests properties of human linguistics and suggests a potential path towards more intelligent natural language processing techniques (Lazaridou et al., 2018).

However, in traditional linguistics, language is studied along different timescales and different levels of deliberation, while recent work on referential games only focuses on long-term language evolution, i.e., modeling how long-term habits develop. In this work, we propose integrated models that consider not only long-term evolution but also short-term equilibrium finding. On the one hand, agents are expected to conform to evolved language habits; on the other hand, in a separate pragmatic stage after the long-term evolution, they are expected to make rational decisions within a particular context so as to communicate more successfully. Our models achieve both of these goals, drawing on psychological game theory (Battigalli et al., 2019), where the payoffs reflect prior beliefs about the strategies, instead of just the final outcome.

**Contributions.** Our key contributions can be summarized as follows:

- We propose a new computational framework that more comprehensively models language dynamics at different timescales. While previous work on referential games considers long-term language evolution (also known as emergentism in linguistics), our model additionally incorporates a subsequent procedure of opponent-aware pragmatic reasoning.

- To this end, we consider action costs of speakers and listeners that can account for several notions of pragmatics, including a game-theoretic pragmatics version that allows us to assess the limits of rational deliberation.
- We conduct a series of experiments that evaluate the relative merits of different pragmatic models, showing that they can improve the empirical communication accuracy both in typical referential game settings (Lazaridou et al., 2018) and in a StarCraft II simulation (Wang et al., 2019b).

## 2 Background and Related work

### 2.1 Linguistic Background

Language system dynamics is studied at several distinct time scales (Lewis, 2014). Cultural language systems mostly emerge and evolve along long time scales and can be considered a steady prior in short term pragmatics, where language strategies of interlocutors reach temporary equilibria.[1]

**Pragmatics.** Pragmatics has the shortest timescale and pertains to conscious rational and cooperative actions of humans (Korta and Perry, 2020). While the term has numerous definitions in linguistics, logic, and philosophy, in general, key aspects include reasoning about the interlocutors' intentions and the ambiguities beyond the expressions, according to the contexts of the conversation (Trask, 1999; Grice, 1975). Computational models have often been influenced by the early work on signaling games (Lewis, 1969) and related incomplete information games theories (Fudenberg and Tirole, 1991). Later works (Parikh, 2001) established more comprehensive models. They mostly considered variants of disambiguation problems, e.g., scalar implicatures (Rothschild, 2013), politeness implicatures (Clark, 2011), irony, debating (Glazer and Rubinstein, 2006), negotiation (Cao et al., 2018), referring expression generation (Orita et al., 2015), and rhetoric phenomena (Kao et al., 2014a,b). More recently, empirical models have been proposed to simulate realistic human pragmatic behavior (Goodman and Frank, 2016; Smith et al., 2013; Khani et al., 2018; Andreas and Klein, 2016; Shen et al., 2019; Achlioptas et al., 2019; Tomlin and Pavlick, 2019; Cohn-Gordon et al., 2018, 2019), revealing practical uses of pragmatics. Zaslavsky et al. (2020) and Wang et al. (2019a) formalized pragmatics models from information and optimization theoretical viewpoints, but did not take game theoretical equilibria into consideration.

**Evolution and emergentism.** Emergence and evolution has the longest timescale and is a more habitual psychological process. In recent years, research on emergent communication (Lazaridou et al., 2018; Wang et al., 2019b; Zaïem and Bennequin, 2019) has regained popularity using multi-agent reinforcement learning (MARL) settings such as referential games. On the one hand, a communication system helps for multi-agent tasks with complex environments and reward mechanisms (Foerster et al., 2016a,b, 2019), especially when the agents can only partially observe the world state. On the other hand, the emergent languages themselves may exhibit functions and characteristics of human languages. For example, when the world state keeps changing and the vocabulary is a limited resource, certain phenomena of language can be observed, such as semantic drift and compositionality (Choi et al., 2018; Lee et al., 2018; Cao et al., 2018; Evtimova et al., 2018; Havrylov and Titov, 2017; Lowe et al., 2019).

### 2.2 Multi-Agent Reinforcement Learning

MARL allows multiple agents to learn coordination strategies in uncertain and dynamic environments, and recently has witnessed vigorous progress in both value-based (Tan, 1993; Rashid et al., 2018) and policy-based (Foerster et al., 2018) methods. Learning language or communication provides effective interaction channels for learning agents, and is an important area of modern AI research (Foerster et al., 2016a; Singh et al., 2019; Das et al., 2019; Lazaridou et al., 2017). Inspired by the theory of mind (TOM) (Goldman, 2012), opponent modeling enables agents to model the anticipated movements of others and holds promise for addressing problems such as non-stationarity (Hernandez-Leal et al., 2017) in a scalable way. One basic method of opponent modeling is policy reconstruction, which predicts action probabilities of a target agent by assuming specific model structures and learning the model parameters based on observed actions. A variant is recursive reasoning, which introduces

human-like thinking by simulating higher-order belief sequences as in *I think that you think that I think...* (Albrecht and Stone, 2018).

While reinforcement learning is widely used in dialogue systems and NLP (Yang et al., 2020), incorporating MARL into NLP is now as well a vibrant and promising avenue (Li and Bowling, 2019; Vered et al., 2019). Concurrently with our work, Lazaridou et al. (2020) explores simple pragmatics for reranking the outputs of generic language models on task-specific conditions, but does not consider game theory. From the viewpoint of MARL, long timescales correspond to stochastic games with random state (historic environment/context) transitions. Short timescales correspond to stage games, i.e., for each state, we regard the game as a stateless game; the target and distractors remain the same for each short-term instance, in which agents' emergent parameters (hereinafter $P_{S_0}$ and $P_{L_0}$) can be seen as steady priors for (multi-round) pragmatic reasoning methods like best responses.

## 3 Pragmatic Models for Referential Games

### 3.1 Long-Term and Short-Term Referential Games

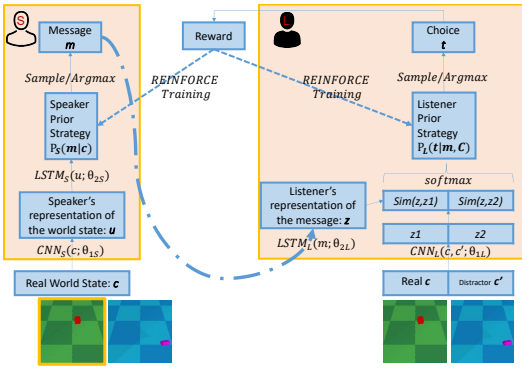

Figure 1: Long-term training & baseline testing.

Inspired by the timescales of language dynamics and their computational frameworks, we define two kinds of referential game scenarios, namely long-term games and short-term games.

In a long-term game setting, we follow the training framework of Lazaridou et al. (2018) to obtain an emergent language system. Figure 1 illustrates the structures of the interlocutors and the training process. For each instance, a candidate set of objects $c_i \in C$ is provided to both agents, but only the speaker knows the true target $c$. The speaker perceives this target using a CNN and obtains a feature vector $u$. The listener similarly obtains feature vectors $z_i$ for each $c_i$. The speaker's LSTM then encodes $u$ into a probability distribution over messages such that a message $m$ can be sampled and sent to the listener, who decodes $m$ into $z$ via its own LSTM and computes the similarity with each $z_i$. A softmax on the similarities produces a distribution for the choice $t$ among the candidates. Both agents obtain a reward $R$, which is 1 if $t$ matches $c$, or 0 otherwise. Based on this reward, they update their CNN and LSTM parameters using the REINFORCE algorithm (Williams, 1992), by maximizing $R \log P_{S_0}(m \mid c)$ and $R \log P_{L_0}(t \mid m, C)$, respectively. In every time-step, if the reward is positive, then the outputs of the agents' policies are encouraged; otherwise, they are discouraged.

After this long-term training, we can regard the emergent networks $P_{S_0}$ for the speaker and $P_{L_0}$ for the listener as fixed (or as slowly-changing) habit priors. We subsequently rely on them as the basis for the following test phase, during which agents may conjecture with regard to each other's habits in order to successfully communicate. However, they cannot rely on pre-defined rules to reach agreement, apart from generic principles such as game theory. The baseline test method, as considered in previous work, is to simply take the arg-max $m$ and $t$ from the two respective prior distributions. However, in this paper, we propose the subsequent methods of refining this.

### 3.2 One-Sided Pragmatic Models

**SampleL.** Andreas and Klein (2016) proposed a framework in which the speaker has a mental model of the listener. Supposing $c$ is the target, the speaker considers several possible messages but finally picks $\operatorname{argmax}_m P_{S_0}(m|c)^\lambda P_{L_0}(c|m)^{1-\lambda}$. Here $\lambda$ represents to what extent the speaker appreciates the message's consistency with their prior habit. The listener then samples $t$ for this message $m$ from $P_{L_0}$.

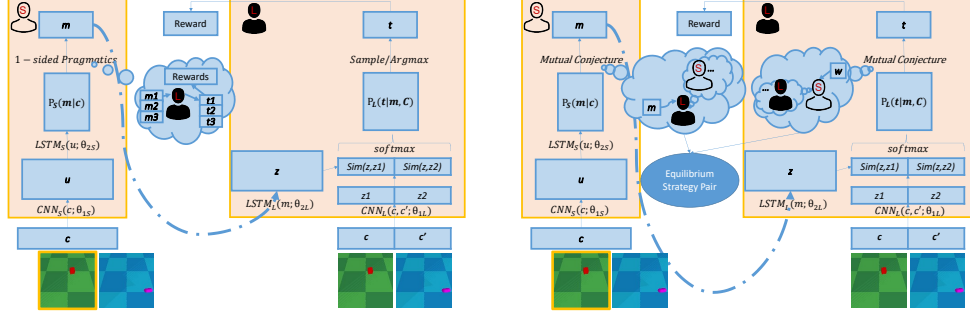

Figure 2: Short-term test using one-sided (left) and two-sided (right) pragmatics.

**ArgmaxL.** A variant of SampleL is when, instead of sampling, the listener simply picks arg-max $t$. In this case, the speaker should pick $\operatorname{argmax}_m P_{S_0}(m|c)$ s.t. $\operatorname{argmax}_t P_{L_0}(t|m) = c$. If none of the messages lead to a correct listener decision, the speaker sends $\operatorname{argmax}_m P_{S_0}(m|c)$.

### 3.3 Two-Sided Pragmatic Models

One-sided pragmatic models equip the speaker with a greater degree of interlocutor awareness, enabling it to account for the listener's thought processes and thereby improve communication accuracy. However, the listener merely follows its pre-existing habits. When the candidate objects are similar, we observe that different high-probability messages tend not to make much difference for the listener's decision. The speaker may have to deviate strongly from its habits to lead the listener to a correct choice.

In the real world, both interlocutors normally have a mental model of each other and both adjust their prior strategies accordingly. There have been a number of mathematical frameworks that model the process of mutual conjectures. In general, starting from $P_{L_0}$ and $P_{S_0}$, one can consider iteratively updating the agent strategies as $P_{S_{k+1}}(m|t) \propto [P_{L_k}(t|m)/\text{cost}(m)]^\alpha$ and $P_{L_{k+1}}(t|m) \propto [P_{S_{k+1}}(m|t)P(t)]^\beta$, for all possible $t \in \{c_i\}$ and related $m$. In our setting, we define the cost of a message as the reciprocal of the speaker's prior. Additionally, $P(t)$ are identical across candidates. The parameters $\alpha$ and $\beta$ reflect the uncertainty of the belief distribution.

**RSA Model.** When $\alpha$ and $\beta$ are small, the new probabilities tend to be evenly distributed. If we set them as 1, we obtain an instance of a Rational Speech Act model (RSA) (Khani et al., 2018; Goodman and Frank, 2016): $P_{S_{k+1}}(m|t) \propto P_{L_k}(t|m)P_{S_0}(m|t)$ and $P_{L_{k+1}}(t|m) \propto P_{S_{k+1}}(m|t)$.

**IBR Model.** If, in contrast, $\alpha$ and $\beta$ are infinitely large, the new probabilities will be a peak distribution and we obtain an instance of an Iterated Best Response model (IBR) (Franke, 2009): $P_{S_{k+1}}(m|t) = \delta[m - \operatorname{argmax}_m P_{L_k}(t|m)P_{S_0}(m|t)]$ and $P_{L_{k+1}}(t|m) = \delta[t - \operatorname{argmax}_t P_{S_{k+1}}(m|t)]$. Under this framework, SampleL ($\lambda = 0.5$) is equivalent to using IBR's $S_1$ and $L_0$ as action strategies.

#### 3.3.1 Psychological Game Theoretic Pragmatics Model

We propose an additional option based on psychological game theory (Battigalli et al., 2019) and the Games of Partial Information (GPI) model (Parikh, 2001), where each agent's actions result directly from game equilibria instead of iterated reasoning. The payoff of a pair of speaker and listener strategies is determined by whether they can bring about successful communication and their consistency with prior language habits, i.e., how "natural" the final actions are. Compared to the other two-sided models, we expect a better integration of long-term priors and short-term rationality under this explicit game theory framework.

**Strategies and payoffs.** Suppose that for a test instance, the candidate set is $C = \{c_1, c_2, ..., c_{|C|}\}$. For each $c_i \in C$, similar to the models above, the speaker proposes a set of messages with non-trivial probabilities $M_i = \{m_{i1}, m_{i2}, ..., m_{i|M_i|}\}$. The set of all possible messages is $M_\cup = \bigcup_{i=1}^{|C|} M_i = \{m_1, m_2, ..., m_{|M_\cup|}\}$. We define the speaker's strategy space, in which each strategy consists of messages to be sent for each candidate and takes the form $s = (m_{s_{c_1}}, m_{s_{c_2}}, ..., m_{s_{c_{|C|}}})$ where

$m_{s_{c_i}} \in M_i$. We likewise define a listener's strategy space, in which each strategy consists of choices of candidates for each proposed message and has the form $l = (c_{l_{m_1}}, c_{l_{m_2}}, ..., c_{l_{m_{|M_\cup|}}})$.

We then consider whether a pair of speaker strategy $s$ and listener strategy $l$, in the respective forms given above, could match if they can bring about accurate communication for each candidate: $\forall i \in \{1, 2, ..., |C|\}, l_{m_{s_{c_i}}} = i$. If they do not match, then $\text{Pay}(s, l) = (0, 0)$; otherwise the payoffs correspond to how consistent their strategies are with language habits, $\text{Pay}(s, l) = (P_S(s), P_L(l))$. Here, the psychological game payoffs are defined as $P_S(s) = \prod_{i=1}^{|C|} P_{S_0}(m_{s_{c_i}} | c_i)$ and $P_L(l) = \prod_{j=1}^{|M_\cup|} P_{L_0}(c_{l_{m_j}} | m_j, C)$. Note that we assume the strategies to be pure ones, instead of mixed distributions. For our game settings, due to the monotonicity relationships between the strategies and their payoffs, pure strategies are sufficient to achieve optimal solutions.

**Equilibrium selection.** We can reasonably adopt the following action selection protocols that can be considered common knowledge shared by the interlocutors without pre-negotiation.

- First build the game payoff table as described above, such that we can then obtain the set of all its Nash equilibria $E = \{(s1, l1), (s2, l2), ...\}$. If the set is empty, the agents act randomly. If there is only one equilibrium, then the agents select it. However, it turns out that when the candidate objects are similar, there tend to be multiple equilibria and the selection is challenging.
- In such circumstances, if there is a Pareto equilibrium, then the agent selects it. If the algorithm stops here, we refer to it as the GameTable model.
- If there is no Pareto equilibrium, it is still possible to determine what a message represents. Note that in such a sequential game, the speaker's message itself contains information about the speaker's strategy. If a message $m$ always corresponds to one specific candidate $c_t$ among all equilibrium speaker strategies that include it, i.e., $\exists m(\forall s \in E(m \in s \to m = m_{s_{c_t}}))$, then its occurrence must represent $c_t$. If $c_t$ happens to be the target, then the speaker can determine to use this message and the listener can understand it. We refer to this as GameTable-sequential.
- For both GameTable and GameTable-sequential, it may occur that the model cannot determine the selection. In this case, the agents randomly select their actions.

The complexity is $O(|M_\cup||C|)$ for RSA/IBR and $O(|M_\cup|^{|C|}|C|^{|M_\cup|})$ for game theoretic models, but there are typically no scalability concerns in our pragmatics context, since the message set size $|M_\cup|$ is usually small. Note that investigating scalability to large $|M_\cup|$ may be important in other natural language cases, and the incremental methods of Cohn-Gordon et al. (2018, 2019) can be applied to game theoretic models in each time step in an *unrolling* procedure, which may make them tractable.

## 4 Experiments

### 4.1 Experimental Setup

We generated MuJoCo-like objects using PyBullet (Coumans and Bai, 2016–2020), with 8 possible colors and 5 possible shapes, at random locations on a color-changing background. The training set consists of 3,000 such objects and the test set contains 1,000. Each training and testing instance has a candidate set of 2 objects. For training, objects are repeatedly sampled as target or distractor from the 3,000 instances, with different position and orientation each time. The long-term training processes 1,000,000 instances. Note that agents have different viewpoints on the target, so that they do not learn to communicate simply by comparing pixel-wise information. Instead, they are expected to learn to encode and decode the object features from individual CNN outputs. In order to reduce the training time, we invoke two separately pretrained AlexNet CNNs. These do not share parameters, so the CNN outputs for the same object will differ between the two agents. To encourage exploring more policies, agent actions are sampled from $P_{S_0}$ and $P_{L_0}$ and these distributions are also penalized by an entropy term if they are not sufficiently evenly distributed at an early stage. Following the setting with the best baseline performance in Lazaridou et al. (2018), messages have a maximum length of 5 symbols and the alphabet size is 17.

For one testing epoch, each of the 1,000 test objects serves as the target once, while distractors in each round are sampled randomly. Similar to Lazaridou et al. (2018), the emergent language system mainly captures color information about the objects, as well as some of the location signals. The messages in Table 3 show the lexicon–object mapping. Virtually identical messages are produced for candidates with the same or even just similar colors (e.g., red and magenta), which makes them

Table 1: Performance of different short-term pragmatic models. SP and LP represent when an instance results in successful communication.

| Test set | Overall | | | Challenge | | |
|---|---|---|---|---|---|---|
| Model | Acc±sd | SP±sd | LP±sd | Acc±sd | SP±sd | LP±sd |
| Original | 85.7 | N/A | N/A | N/A | N/A | N/A |
| Baseline | 90.1±0.9 | 0.53±0.00 | 0.97±0.00 | 53.0±2.6 | 0.56±0.02 | 0.80±0.01 |
| SampleL$\lambda$0 | 90.1±1.1 | 0.26±0.00 | 0.97±0.01 | 54.8±2.9 | 0.23±0.02 | 0.79±0.01 |
| SampleL$\lambda$0.5 | 89.2±1.1 | 0.53±0.00 | 0.97±0.01 | 53.9±4.5 | 0.55±0.02 | 0.77±0.01 |
| ArgmaxL | 90.6±0.9 | 0.53±0.00 | 0.97±0.00 | 56.6±2.5 | 0.53±0.02 | 0.79±0.01 |
| RSA_2rnd | 91.9±0.9 | 0.53±0.00 | 0.95±0.00 | 55.9±1.9 | 0.54±0.01 | 0.66±0.05 |
| IBR_2rnd | 95.5±0.6 | 0.51±0.00 | 0.96±0.01 | 80.6±2.8 | 0.43±0.02 | 0.78±0.01 |
| RSA_cnvg | 93.1±0.8 | 0.53±0.00 | 0.95±0.01 | 62.0±2.1 | 0.54±0.01 | 0.67±0.05 |
| IBR_cnvg | 95.6±0.5 | 0.51±0.00 | 0.96±0.01 | 80.6±2.8 | 0.43±0.02 | 0.78±0.01 |
| GameTable | 94.9±0.6 | 0.51±0.00 | 0.95±0.00 | 74.6±2.0 | 0.47±0.01 | 0.72±0.03 |
| GameTable-s | 98.8±0.1 | 0.49±0.00 | 0.93±0.01 | 94.0±0.6 | 0.36±0.01 | 0.68±0.03 |

hard to disambiguate and cause most of the false predictions of the baseline. To show how pragmatic models help in such cases, in addition to the original overall test set, we separately pay attention to a challenging subset of around 200 instances, where the candidate's colors are the same or similar. We run the overall test set and the challenging set for 5 epochs each for every kind of pragmatic model and record the accuracy metrics in Table 1. We also care about the consistency of the pragmatic actions with the long-term priors, assessed by computing prior probabilities of speaker messages and listener choices for the instances with successful communication, recording them as the SP and LP values in Table 1. All the pragmatic models involve generating message proposal sets for objects. We take the highest probability messages that sum up to 75% and filter out the trivial long tail ones. On AWS t3.xlarge (4 CPU 16G Memory), the training takes about 1 day and the total time to test all methods takes about 1 hour.

## 4.2 Results

**Baselines.** Given the results in Table 1, an initial observation is that our baseline communication accuracy exceeds that of Lazaridou et al. (2018). This may stem from our use of pretrained CNNs, while the original paper trained them during the long-term game. While this baseline language system approaches the limit of training performance, on the challenging disambiguation test set, however, the baseline accuracy drops to nearly 50%, indicating that similarly colored candidates cannot be distinguished using literal language meanings. This is similar to ambiguity issues in natural language (de Melo et al., 2012; Li et al., 2017), including also scalar implicatures.

**One-sided pragmatics.** One-sided pragmatics models are also regarded as benchmarks. The SampleL results suggest that one-sided pragmatics models succeed when the speaker considers the listener's model carefully enough. However, the improvements in communication accuracy are not substantial. For SampleL with $\lambda = 0$, the improvement is moreover achieved at the cost of significantly violating the speaker's prior beliefs. We conjecture that when the candidates are similar, their proposed messages are similar, and the listener generates similar $P_{L_0}(t|m)$ for each message. Thus, few proposed messages ultimately lead to a major difference in the listener's choice. In fact, we found that the speaker sometimes sent useful new messages, but the listener was not sufficiently sensitive to recognize them. This suggests that two-sided models may be favorable.

**RSA and IBR.** We considered hierarchy depths of 2 ($S_0 - L0 - S1 - L1 - S2$) to model a human-like bounded rationality, as well as unlimited depth hierarchies until strategy convergence to assess its potential limits. Table 1 manifests that the 2-round IBR model reaches convergence. This suggests that a human-like bounded rationality may be sufficient in this task scenario. RSA has a low performance, which implies that for this game, deterministic rationality proves advantageous over probabilistic decision making. According to Zaslavsky et al. (2020), while keeping $\beta = 1$, each recursion step $k$ amounts to maximizing $\alpha\mathbb{E}_{c,m|c\sim P_{S_k}}[\log P_{L_k}(c|m) + \log P_{S_0}(m|c)] + H_{P_{S_k}}(M|C)$. Here, the first item is the expectation of speaker utilities (corresponding to Acc and SP) over the object candidates and the speaker messages conditioned on them. The second item is the conditional entropy of the

Table 2: Pragmatics communication accuracy using virtual interlocutors.

| Virtual Opponent | S-Fide | L-Fide | ArgmaxL | RSA | IBR | GT | GTs |
|---|---|---|---|---|---|---|---|
| Exact copy | 100.0% | 100.0% | 56.6±2.5 | 62.0±2.1 | 80.6±2.8 | 74.6±2.0 | 94.0±0.6 |
| Training 100k rnd | 96.6% | 97.0% | 53.0±2.3 | 54.3±1.4 | 68.6±1.6 | 58.1±2.2 | 69.9±1.8 |

messages: a larger entropy means less precise messages. For RSA ($\alpha = 1$), the first item has less importance, so we obtain a low performance and the decisions are probabilistic. For larger $\alpha$, the performance improves and the decisions are deterministic. This may explain why IBR performs better.

**IBR and GameTable.** IBR and GameTable are two ways to achieve an equilibrium state of mutual conjectures. Hence, it is natural to find that they obtain similarly high accuracy on the overall test set. For the challenge test set, IBR has a better accuracy than GameTable, but a lower speaker payoff. The reason is that the IBR speaker and listener always set their strategies as the best response according to each other, so the amount of irrational choices is minimized. In comparison, GameTable is not guaranteed to have a Pareto optimal equilibrium, leading to a slightly lower accuracy. At the same time, the communication success and the consistency with the prior beliefs are explicitly specified in GameTable's payoffs, so if there is a Pareto equilibrium, then both are guaranteed. In contrast, the iterated calculations in IBR sometimes diverge from the initial priors, considering that best responses are a form of non-linear transformation. Some potentially optimal actions are set directly to zero in initial rounds because they are not the best choice at that time.

**GameTable-sequential.** Among all the frameworks, GameTable-sequential always obtains the highest accuracy. Since agents cannot find a Pareto solution to handle all scenarios, they decide to prefer communication success while sacrificing consistency. The game is then less about achieving a Gricean conversation and more about just achieving a consensus, so we observe a low speaker payoff. Thus, GameTable-sequential serves as an upper bound for pragmatic communication accuracy under the short-term game assumptions.

### 4.3 Robustness for Virtual Opponent Models

In classic pragmatic frameworks, it is often assumed that interlocutors know about each other very well. For example, the prior probabilites $P_{S_0}$ and $P_{L_0}$ are considered shared knowledge. However, in practice, game information may be incomplete and one's assumptions about the other interlocutor may diverge from reality. Each person learns their own language model and reasons about others based on their own mental model. To check how this affects pragmatics, we assume agents $S$ and $L$ first respectively learn to speak and listen as before, so we obtain $P_{S_0}$ and $P_{L_0}$, $S$ has its own listener model $P_{L_0'}$ and $L$ has its own speaker model $P_{S_0'}$. We then train $P_{S_0'}$ and $P_{L_0'}$ by communicating with and simulating outputs of $P_{S_0}$ and $P_{L_0}$ regarding game instances. In the end, $P_{S_0'}$ is similar to $P_{S_0}$ but not exactly the same, and similarly for the listener model. This amounts to policy reconstruction methods from the perspective of theory of mind and opponent modeling. During testing, $S$ and $L$ try to communicate using pragmatic frameworks, each using their own model as the virtual interlocutor model. The results for this are given in Table 2. "Fide" here describes the average fidelity when a virtual model simulates the real one, which is computed as the cosine similarity between the output distributions of the models on the same input. We observe that although the fidelities are fairly high after training, their minor differences substantially hamper the pragmatic performance. GameTable is most susceptible to this, while IBR and GameTable-sequential are fairly robust.

### 4.4 Language Inspection

Table 3 inspects the messages. For this, we traversed the test set and obtained all possible messages and their significant corresponding target features (color and location). After the long-term training, the emergent language system reveals potential characteristics of compositionality. Table 3 shows that the first two digits (prefixes) of the messages principally relate to color, as well as occasionally some location information. The third digits (suffixes) reveal key location information. Here, a plain font represents mappings found using the baseline framework, while bold font represents the new mappings found using the game-theoretic pragmatic framework, in which some location and color

Table 3: Illustration of the main lexicons that occurred. Baseline lexicons in plain font. New lexicons from GameTable-sequential in bold font. The backgrounds are removed to make images overlap better.

| okccc | **okdcc** | **okncc** | dkccc | **dkhcc** | dodcc | **dcccc** | dnhcc | dnccc | nidcc | **niccc** | nkccc | **nkicc** |
|---|---|---|---|---|---|---|---|---|---|---|---|---|

features become distinguishable, thus aiding in the more challenging disambiguation tasks. For example, after baseline training, almost all yellow and white targets are expressed by the speaker as *okccc*. After the pragmatics computation, it expresses some white targets as *okdcc* and some right hand side yellow objects as *okncc*. In essence, these new messages and their unique meanings are explored and trained during the long-term training phase. However, they provide little benefit when using the baseline or one-sided framework. Only the two-sided frameworks are able to discover the feasibility of exploiting these potential messages.

## 5 Case Study: Pragmatics between StarCraft II Allies

For further analysis, we consider an additional setting based on the StarCraft game series. StarCraft unit micromanagement involves fine-grained control of individual units and their communication has recently attracted substantial research interest due to its high degree of control complexity and environmental stochasticity. A recent work called NDQ (Wang et al., 2019b) provides a framework for centralized training with decentralized execution for allies in StarCraft II games[2]. Each ally agent is modelled by a deep recurrent Q-network (DRQN), which takes in its local observation–action history and emits action–value pairs at each time step. Since the battle result serves as global feedback for all allies, NDQ trains a mixer network and assigns appropriate credit for each ally at the training phase. A unique aspect of this work is that agents also learn to communicate with each other using succinct and expressive messages. During the training process, agents gradually learn to drop messages that cannot reduce the uncertainty in the decision-making process of other agents.

In NDQ, at time step $t$, agent $i$ encodes its action–observation history $\tau_i^t$ as a real-valued embedding $f(\tau_i^t)$. To send it to others, the agent samples a raw message $s_r(f) \sim \mathcal{N}(f, \sigma I)$. The agent then drops useless bits in the raw message to form a succinct message $s(f)$. The process of generating $s(f)$ from $\tau$ can be regarded as the speaker process. Specifically, NDQ uses specially designed loss functions to force useless bits to be near the origin of the message encoding space, so that the distance from the origin is used as the dropping standard deciding which bits cannot reduce the uncertainty of decisions of other agents. In the listener process, other agents directly receive the trimmed message and feed it into their DRQNs. In our work, we attempt to use pragmatics to further improve the message succinctness and resist message drop. We notice that $f(\tau_i^t)$ usually does not deviate substantially from $f(\tau_i^{t-1})$, so it is reasonable to assume $f(\tau_i^t) \sim \mathcal{N}(f(\tau_i^{t-1}), I)$. Moreover, the initial configurations in all episodes are similar in a StarCraft Multi-Agent Challenge map, so $f(\tau_i^0) \sim \mathcal{N}(\mu_i, I)$, where $\mu_i$ represents the average embedding value of agent $i$ at time 0. This kind of prior distribution amounts to the limited candidate set in short-term referential game settings, and we can adjust both sides, i.e., the speaker and listener strategies, based on the following principles.

(1) If there is no message drop, the listener $l$ should reconstruct $f$ from $s$, so $\forall f, l(s(f)) = f$, which is a requirement for both the speaker and the listener. (2) To better resist message drop, the speaker should seek to reduce the amount of information of $s(f)$ and the expected loss of element values within it, which motivates minimizing $-H(s(f)) + \int_f p(f)||s(f)||_1 \, df$. Note for each pair of speaker and listener strategy, linear scaling of the message values yields an infinite number of equivalent

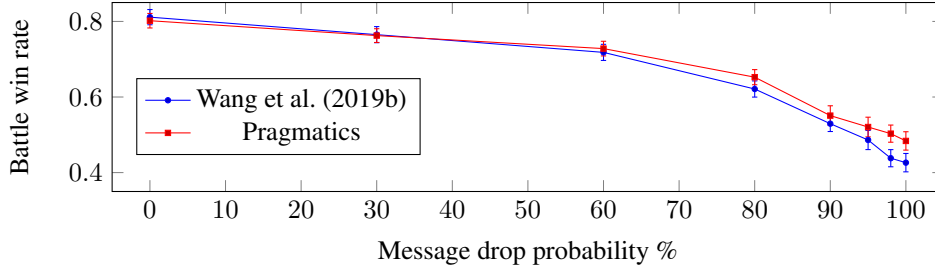

Figure 3: StarCraft II resistance against message dropping. Tested on the SC2 map `1o2r_vs_4r`, featuring 1 Overseer and 2 Roaches fight against 4 Reapers.

solutions, so we constrain $\det(\text{cov}(s(f)))$ to 1. Then it is easy to find an equilibrium solution $s(f) = f - E(f)$ and $l(s(f)) = s(f) + E(f)$. This means $s(f)$ is white Gaussian noise with the largest entropy and smallest $L_1$-norm given $\det(\text{cov}(s(f))) = 1$. Note that in this game setting, we focus on the improvement regarding succinctness brought by pragmatics, so the consistency with prior language habits is not important. In addition, agents do not need to bother with equilibrium selection, since they can easily agree using this simple pre-defined equilibrium. Our overall game setting and the agent networks are exactly the same as in Wang et al. (2019b).

We ran 30 million training steps and update the model by sampling 8 episodes from a replay buffer in each step. The training was conducted on an AWS g4dn.xlarge GPU (NVIDIA T4 Tensor Core GPU) instance and took about 20 hours. For evaluation, we drop message bits randomly and average battle win rates under different drop rates over 100 random seeds. 16 episodes are tested for each seed. In Figure 3, we present the performance comparison on map `1o2r_vs_4r`, along with $95\%$ confidence intervals. We observe that, as the dropping rate increases, pragmatics shows its advantage. We also obtain a form of division of labour. For example, at time step 0, the message (0,0,0) corresponds to different pragmatic meanings for different agents.

## 6    Conclusion

This paper proposes novel computational models incorporating opponent-aware short-term pragmatics reasoning into long-term emergent language systems, inspired by interdisciplinary work in multi-agent reinforcement learning and computational linguistics. The experimental results suggest that under our referential game settings, two-sided pragmatics models outperform one-sided ones by finding ways to exploit the potential of the language to a greater extent. Empirically speaking, IBR achieves significant communication accuracy with a practically viable procedure for humans, where typically two-level reasoning is sufficient. This accords with the conclusions from cognitive science, though IBR occasionally fails with sub-optimal actions. Our advanced game-theoretic model provides new upper limits for communication accuracy, outperforming the classic model. We further show that the model can be applied to communication between StarCraft II allies, allowing them to communicate more efficiently and thereby mitigating the effects of message drop. Our novel model shows robustness under challenging settings, opening up important paths for future research. Our code is freely available online. [3]

## Broader Impact

This paper studies fundamental principles of communication, attempting to shed light on conditions under which pragmatic reasoning can enable agents to communicate more efficiently in challenging circumstances. Our interdisciplinary work draws novel connections between several distinct branches of linguistics, as well as multi-agent reinforcement learning and game theory. We believe it is important to develop models of agents endowed with communicative abilities that not only allow for the long-term emergence of linguistic patterns but also incorporate interlocutor awareness (theory of mind) and game theory. Recent work using language to achieve common objectives and cognitive studies show that feedback is paramount for language learning (Zaïem and Bennequin, 2019). Thus,

these sorts of models could be important for studying how to go beyond regular data-driven training of (neural) language models. In the long run, this sort of research has the potential to enable chatbots or other forms of virtual personal assistants endowed with more empathy and better awareness of the state of mind of the person they are interacting with. This might also allow them to better adapt to the needs of underrepresented social groups and different personalities.

## Acknowledgements

We wish to thank Prof. Zihe Wang for inspiring discussions.

## Footnotes

[1]Of course, in reality, the cached equilibria from repeated interactions also conversely give rise to changes on the long term timescale, but we leave this for future work.

[2]We use the setup introduced by Samvelyan et al. (2019). We consider combat scenarios where the enemy units are controlled by StarCraft II's built-in AI (at difficulty level 7), and each of the ally units is controlled by a learning agent. At each time step, each agent chooses one action from a discrete action space consisting of the following actions: noop, move[direction], attack[enemy_id], and stop. Under the control of these actions, agents move and attack in a continuous state space. A global reward that is equal to the total damage dealt on the enemy units is given at each timestep. Killing each enemy unit and winning a combat entail extra bonuses of 10 and 200, respectively.

[3]`https://fringsoo.github.io/pragmatic_in2_emergent_papersite/`

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
