[Reviews · NeurIPS 2020]

Review 1

Summary and Contributions: This paper proposes to apply pragmatic reasoning methods, which uses speaker and listener agents that recursively model each other, to improve the communicative accuracy of speakers and listeners in an emergent communication reference game setting. The paper presents a method for pragmatic reasoning that solves for game strategies that maximize a combination of task success and the probability of the messages and selected objects under the speaker and listener distributions, respectively. The paper finds that the proposed game-based approach obtains higher task success than various recursive models of pragmatic reasoning in a object description reference game where the listener has a perfect model of the speaker and vice versa, as well as a setting where the listener and speaker use distilled versions of each other. Finally, the paper presents results on a form of pragmatics in multi-agent Starcraft game setting.

Strengths: Combining pragmatic reasoning with emergent communication is well-motivated, and will likely be of interest to a range of researchers in computational pragmatics and multi-agent learning. The main strengths of the contributions are: 1) Proposing a method for pragmatic reasoning based on exact computation of game equilibria, and comparing this to a wide range of layered reasoning methods from past work. 2) Applying all of these pragmatic reasoning to models trained in an emergent communication setting, and showing that pragmatic reasoning improves communicative accuracy. 3) Showing that a pragmatic method obtains higher performance than a method from past work on multi-agent communication in a Starcraft game setting, when messages have a high probability of being dropped. The experiments on using "virtual interlocutors" in Table 2 (not having exact access to the listener model) were also valuable (and more emphasis should be placed on them).

Weaknesses: Both general lines of work in this paper, on emergent language to learn speakers and listeners, and on explicit pragmatic reasoning to improve communicative success or efficiency, have been explored in prior work. Their integration here was limited to using speakers and listeners learned through emergent communication as the base models in pragmatic reasoning. I appreciate the approaches the paper uses, but I felt that its aims were currently ill-defined, and the contributions were too spread over several areas so that each of them is thin or unclear. For each of the contributions above, 1) Since, as the paper points out, the GameTable-sequential is essentially an upper bound on communicative accuracy, the results in 4.2 are largely useful for analyzing the relative merits of the methods, but details about the proposed game equilbria method, and the comparisons to past work, were unclear. See Clarity. 2) It was unclear why communicative accuracy (task success) is a useful primary evaluation metric in the emergent communication framework, particularly when the reasoning procedure used by the listener has perfect access to the speaker's responses. Emergent communication gives a framework to investigate compositionality or efficiency of language, or concept formation, and the qualitative analysis in 4.4 is a partial step toward this, but it's unclear how representative these examples are. To strengthen this contribution, could a metric for evaluating compositionality, e.g. topographic similarity (Brighton & Kirby 2006) or tree reconstruction error (Andreas 2019), be used here? 3) The Starcraft section was very compressed, and the method seemed different enough from the rest of the paper that it was difficult to tell what was done here (see Clarity). How artificial is the message dropping (the only setting where pragmatics produces a benefit) in comparison to the standard message dropping (line 294)? Brighton & Kirby 2006: Understanding linguistic evolution by visualizing the emergence of topographic mappings Andreas, 2019: Measuring Compositionality in Representation Learning

Correctness: The claim in line 35, "a multi-agent policy gradient reinforcement learning framework... including a game-theoretic pragmatics version" was misleading, since if I understand correctly no pragmatic reasoning is incorporated into the policy gradient updates (the models are trained only on communicative accuracy). Are the same sets of message candidates used in all pragmatic methods in Table 1 (GameTable, GameTable-s, and RSA, IBR, etc.)? This is important since the performance of RSA methods depends crucial on the candidate set size. From line 215 it seems likely that all methods do use the same candidates, but the use of "non-trivial" probabilities in line 157 makes it unclear -- does non-trivial mean non-zero, or above the 75% threshold? The RSA methods could likely be improved if the final listener decision takes an argmax from the derived listener model, rather than sampling (as could SampleL): i.e.. the listener is modeled as being noisy, but actually makes deterministic decisions. This would also be in line with past work on RSA listeners. "There are typically no scalability concerns in pragmatics contexts". Is this specific to the particular emergent communication setting investigated (with limited message length and small vocabulary)? A line of work on RSA pragmatic models (e.g. Cohn-Gordon et al. 2018) has been necessary to deal with scalability concerns in realistic linguistic settings where it's uncommon to be able to enumerate the full set of possible messages. Cohn-Gordon et al. 2018, Pragmatically Informative Image Captioning with Character-Level Inference

Clarity: "Incorporating pragmatics into emergent language" was (in my opinion) misleading: If I understand, this work uses a two-stage procedure: first training agents in an emergent communication setting (following Lazaridou et al. 2018) without pragmatic reasoning, and then applying different types of pragmatic reasoning to the learned agents. It may be desirable to separate the two stages of training as this paper does (in contrast to Tomlin and Pavlick 2019, which trains through the pragmatic inference procedure in an emergent communication setting), but this should be more thoroughly discussed and the claims tempered (e.g. line 328, incorporating opponent-aware reasoning into long-term policy gradient). The description of the GameTable method had some details that were unclear: - Is the candidate set C in 3.3.1 the set of candidates for a single instance (i.e. a target and distractor), or is it a global candidate set spanning multiple examples? If global, is it defined using the training set or the test set? - How are the equilibria computed in lines 171-173? - If there is no Pareto equilibrium, and no message m that meets the satisfies the existential in line 181, what message is chosen by the speaker? Separate results are given for GameTable and GameTable-sequential in Table 1, but from the description at the end of Section 3, it seems that GameTable applies to examples with a Pareto equilibrium, and GameTable-sequential to examples with no Pareto equilibrium. Are the examples the two methods are evaluated on in Table 1 disjoint? The method described in section 5, for the Starcraft domain, is substantially different from the previously-described method: it does not use discrete messages, uses a different objective function, and does not use prior probabilities from the speaker or listener models. I felt that this section should be substantially expanded to make it clearer what's done, and the relation to the prior method, if it's retained.

Relation to Prior Work: There are lots of helpful references to relevant work on computational pragmatics and emergent communication, but it's not explicitly stated how this work relates to or combines these. Citations, or more explanation, should be given for the following claims: - line 44-46 about emergent language systems being a steady prior for temporary equilibria of short term pragmatics? - 47: pragmatics involves the most conscious intellectual processes Although I don't think this paper should be penalized for not comparing to them, some other related work discussing how pragmatic reasoning approximates game equilibria are: Yuan et al., 2018. Understanding the Rational Speech Act model Wang et al. 2019. A mathematical theory of cooperative communication Zaslavsky et al. 2020. A rate-distortion view of human pragmatic reasoning

Reproducibility: Yes

Additional Feedback: 84-86: short timescales correspond to stage (stateless) games: does this refer to a single-turn interaction between a speaker and a listener where no parameters are updated? 109: technically, REINFORCE maximizes the expected R, not these quantities (although I see where these came from given the form of the score function estimator) - A citation for REINFORCE should be given, e.g. Williams 1992 120: It's unclear to me why SampleL samples from the listener model, rather than argmaxing, as it seems that argmaxing should obtain better performance. (And Andreas and Klein 2017 did not sample from a listener model [although they did sample from the speaker model], but rather used human listeners to interpret the pragmatic utterances.) 147: It's unclear how IBR is equivalent to SampleL, since SampleL samples from both distributions, so it can produce messages which are not equal to the argmax (while IBR places all probability mass on the argmax). Should this instead be that IBR is equivalent to a variant of ArgmaxL that uses a 0.5 lambda-weighted interpolation of S_0 and L0, or equivalently replaces the sampling in SampleL with argmax? 166: The payoff seems to be all-or-nothing: if the strategies do not match on every candidate output, the payoff is zero. Do all the settings explored here have feasible matching strategies? 301: this assumption seems strong -- does it mean that the feature vector does not condition on the observation and action history at any point? Minor points / additional things to clarify if space: - Does the REINFORCE training use a baseline? 113: "fixed or as slowly-changing habit priors": which are they in these experiments? it seems that they are fixed, but it's not totally clear. 120: how are these candidate messages obtained, e.g. by sampling or by beam search? 138: what is the speaker's prior? from line 144 it seems that this is P_S_k(m | t), but this should be more explicit. 168: should the sum of similarities be exponentiated (since the similarities are softmaxed)? 251: is "consistency" consistency with the speaker prior? Table 2: clarify that RSA and IBR are at convergence. Figure 3: drooping -> dropping ---- update after author response ---- Thanks to the authors for the helpful, and thorough, response! It's much appreciated. I've updated my score to a 6 (from a 4) after the response, rereading parts of the paper, and some discussion with the other reviewers. I feel overall positive now about the paper -- there are a number of interesting threads here that other work will be able to build on. I do still feel the paper is spread a bit thin, and that the framing could be improved, and the clarity could be much improved (extra space will help for this), but I'm more confident about the correctness given the response. To specific points from the response: R1.1: I do still feel that the "integration" is ambiguous, and one interpretation of it is potentially misleading. I'd appreciate if the authors could make it more clear minimally in line 35 ("a multi-agent policy gradient reinforcement learning framework with action costs that can account for several notions of pragmatics") that the pragmatic reasoning procedures are applied in a separate stage *after* the models are trained with multi-agent RL (if I understand correctly). 1) GameTable-s methods: I think there are two separate issues here: how the payoff is defined, and whether an equilibrium is exactly computed. IBR and RSA methods, when iterated, converge to an equilibrium in some settings, implicitly optimizing some game-theoretic payoff (as investigated by Zaslavsky et al.). I don't think this prior detracts from this work! It's nice to see a comparison of an optimal method with a certain payoff against approximate methods (possibly with a different payoff), with "one-sided" being an extreme form of approximation. A more thorough comparison of these two different factors (the payoff being optimized, and how closely the equilibrium is approximated) could be interesting, but beyond the scope of this paper. Thanks for the clarifications on the method, these address my concerns. 2) I appreciate that the methods don't require pre-negotiating a custom protocol, but the perfect opponent models still seems a very strong assumption, and distillation seems to be a relatively weak form of relaxation. But I'm not able to think of a better alternative -- it seems that training models from scratch on a separate subset of the training data would likely result in different emergent languages. I agree compositionality could be interesting to analyze but might be better left for future work. I'm not sure if these topological similarities are significant or even substantial (what are the absolute scores these raises correspond to)? I'm still unconvinced that these improvements, even in StarCraft, demonstrate practical relevance. To make this more convincing, it would help to more clearly convey the strength of the baseline approach in the reference game test set, and give a much clearer description of the StarCraft setting, in particular focusing on conveying how artificial or natural the message dropping might be. (But, I do feel that the work is interesting even if it doesn't demonstrate practical relevance.) R1.4 Time scales: I appreciate the Lewis et al. claims, and conventionalization more generally (e.g. Garrod & Doherty 1994). But if I'm understanding the setup here, it's the reverse of these works: the short-term pragmatic interactions do not inform the long-term habits; rather, pragmatic inference is applied after the long-term habits are developed. It would make the framing stronger if the language and claims around this were made more precise.


Review 2

Summary and Contributions: This paper proposes a new framework that models language learning in communicative agents at different timescales during the language evolution/learning process. They use insights from game theory (determining a Pareto equilibrium based on agents strategy/payoff matrix over the game) and use this for message selection in communicative environments. Experiments in both a simple reference game and a more complex multi-agent domain (as compared against a range of other frameworks that have been extensively used in the EmeComm literature) show that this framework has superior performance in a simple reference game setting. Update: My score was initially positive, and I will leave as is. I thought the review discussion was especially helpful (uncovered lots of small negative points about the paper) but I think overall this still warrants an accept.

Strengths: 1. Some of the results give nice insights into how deterministic rationality might be better than probabilistic decision making/combining prior beliefs, which is interesting to see (and a helpful result for future work). 2. The paper is well written and clearly explained. The figures also give a nice overview of previous frameworks (e.g., probabilistic inference/belief combinations). 3. The multi-agent reinforcement learning experiment (i.e., Starcraft) is nice to see, given that it is a harder domain/task to master than a simple reference game. However, the results are somewhat inconclusive (but are much stronger in the reference game setting). If the authors could also (a) evaluate the policies and (b) test on different splits of complexity, it would make the results a lot clearer. 4. The analysis of time complexity is helpful (however, it would be good to see how practical/scalable this is for different games and different sets of messages)

Weaknesses: 1. Recent work in multi-agent communication (https://arxiv.org/abs/2005.07064) has moved past symbolic and uninterpretable communication channels to allow natural language communication. This is not only more interpretable and allows human intervention, but also allows probing of all of the phenomena mentioned here (pragmatics, semantic drift, evolution of language) etc. Given that the field is starting to move in that direction (and that is where we hope to see more intelligent and robust behaviour from communicative agents) it would be good to move this work from symbolic/discrete communication channels to natural language, to assess if the same hypotheses hold. 2. To add to the above, even though the results show that certain frameworks work better (e.g., game-theoretic-sequential is better than RSA/IBR) it remains to be seen if (a) this hold when we want to move to real natural language i.e., the communicative outputs are semantically/syntactically correct and pragmatically relevant and (b) if the game-theoretic framework is even tractable under those conditions. 3. There are some missing references/ties to existing work. For instance, incremental RSA (chttps://scholarworks.umass.edu/cgi/viewcontent.cgi?article=1045&context=scil) and multi-agent frameworks with natural language (https://arxiv.org/abs/2005.07064) that _are_ jointly learned with a cooperative reward as opposed to RSA 4. This is not particularly important, but there are lots of modeling decisions made that seem somewhat arbitrary (e.g,. using AlexNet, when most would rely on ResNet/Inception and other architectural choices). It would be good to have a small note (or comparison) explaining why. 5. It would be good to add more detail about the time complexity and succintly state when/where this becomes intractable.

Correctness: The experimental setup is sound and explained in detail, however I would add more detail/experiments that show the tractability.. in terms of time complexity of the game theoretic approach.

Clarity: The paper is written well and all experimental details and tables and figures are adequately explained.

Relation to Prior Work: This is well-positioned in the literature; the only missing references that are especially important were released/accepted after the time of submission, so authors should not be held against that. However, it would greatly help this paper if they use insights from those!

Reproducibility: Yes

Additional Feedback:


Review 3

Summary and Contributions: - The paper proposes to combine work from emergent communication + multi-agent RL with pragmatic reasoning - The paper proposes specific test-time augmentations to the usual emergent communication setting motivated by pragmatic reasoning and game theory, using a model of the other agent’s behavior to - The paper tests these augmentations in a referential game and a Starcraft environment, showing improved communicative success

Strengths: - The paper is well-written and clearly motivated. - The idea to combine pragmatics with emergent communication in multi-agent RL is really interesting. The paper is well-executed and thorough, studying and comparing several pragmatic models. - I appreciated the paper’s experiments on the agents learning the models of the other agent (which seems important for this method to have practical applicability), and the qualitative analysis of the messages.

Weaknesses: - As someone without a linguistics background, it was at times difficult for me to follow some parts of the paper. For example, it’s not clear to me why we care about the speaker payoff and listener payoff (separate from listener accuracy), rather than just a means to obtain higher accuracy --- is it important that the behavior of the speaker at test time stay close to its behavior during training? - I think more emphasis could be placed on the fact that the proposed methods require the speaker to have a model (in fact, in most of the experiments it’s an exact model) of the listener’s conditional probability p(t|m), and vice-versa. - I would have liked more description of the Starcraft environment (potentially in an Appendix?)

Correctness: Yes, as far as I can tell.

Clarity: This is a strong point of the paper, with a noted caveat mentioned in the weaknesses above.

Relation to Prior Work: The related work is clearly discussed.

Reproducibility: Yes

Additional Feedback: Overall, I think this is a good paper and I’d recommend acceptance. Other comments: - Should the citation style be numbered, e.g. [1]? - Fig 3 caption: drooping -> dropping


Review 4

Summary and Contributions: 1) The authors expanded previous models to play two-player referential games to more settings. While previous baselines mostly only consider point estimate for the speaker and listener (argmax m and t in Line 117), the authors considered many more scenarios. 2) Through studying each scenarios, the authors concluded that the two-sided pragmatic model are able to both yield better outcomes in the evaluation games and give more intrepretable results, confirming previous findings in psycholinguistic studies.

Strengths: 1) The authors’ method to consider both short-term pragmatics and model long-term emergentism is both novel and theoretically sounds 2) The authors studied various cases under the multi-agent RL framework and conducted extensive studies. 3) The authors presented many interesting comparisons with previous works. The discussion with respect to previous classic pragmatic framework seems particularly insightful. 4) For evaluation, the authors constructed a challenging subset of instances where one defining attribute stays constant (colour of object). I find this experiment informative

Weaknesses: 1) In Line 196, I think the authors could provide more details as to how different the two “distinct” AlexNet are. 2) Perhaps due to a lack of relevant background, I find the discussion between RSA and IBR confusing. Particularly, I would like the authors to explain more on the “hierarchy depths” concept

Correctness: Yes, to the best of my knowledge

Clarity: Yes

Relation to Prior Work: Yes

Reproducibility: Yes

Additional Feedback:

[Author Response · NeurIPS 2020]

**All Reviewers:** Many thanks for the insightful reviews. Please see our responses below (R$n$ refers to Reviewer $n$).

**R1.1 Aim of the paper.** We show that short-term pragmatics can optimize communication beyond long-term emergent habits, with ref.-games and SCII as two cases exemplifying this same core idea. While, recently, Tomlin et al. 2019 and Lazaridou et al. 2020 also consider pragmatics during training, they do not use two-sided adjustment for each specific test instance, thus neglecting potential improvements. As in the early literature on emergent communication, although the invoked settings are artificial, the findings may inspire similar attempts at modeling natural language phenomena.
**(1) GameTable(-s) methods.** Comparing to the numerous RSA/IBR frameworks, using habit consistency as a game theoretical payoff is a novelty, to the best of our knowledge. Recent work (Zaslavsky et al. 2020, Wang et al. 2019) formalized RSA from information and optimization theoretical viewpoints, but did not take game theoretical equilibria into consideration. Details: Object candidate set $C$ is for a single instance. Equilibria are simply extracted from the table according to their game theoretical definition. GameTable only checks Pareto. GameTable-s checks Pareto first; only if there's no Pareto, it goes on to the following procedures. These two methods are compared on the same test instances. Line 166: It may occur that there is no equilibrium or there are multiple equilibria without Pareto/sequential solution. In this case, the agents randomly select their actions.
**(2) Accuracy and compositionality metrics.** Accuracy improvements may seem easily obtainable when assuming agents have perfect opponent models. However, note that agents cannot pre-negotiate any custom protocol. Rather, we make very mild assumptions: RSA / IBR / GameTable are all universally-known and widely used principles for achieving consensus. How much accuracy these methods can bring about is worth exploring and similar improvements in natural language may be possible. Besides, communication accuracy in StarCraft-II shows the practical relevance of such settings. Compositionality is an important metric and we may add the following to Sec. 4.4: Topological similarities (w.r.t. location and color features) rise by -0.013, 0.213, 0.030, 0.035, 0.029, 0.062, respectively for black, blue, green, cyan, red/magenta, yellow/white objects in the challenging test set when using GameTable-s instead of the baseline method.
**(3) Starcraft-II.** Indeed, we adapted the reasoning procedure to fit the setting, and we emphasize succinctness and robustness, rather than compositionality or consistency; however, note that these are as well important metrics of any communication system. Details: Line 294 means agents learn to emit informative messages in long-term training; for short-term tests, baselines and our methods are compared with the same drop probabilities. Line 301: Feature vectors do condition on observation-action, but observations themselves (e.g., allies' location and health) do not deviate much from the last time step; which is why feature vectors do the same. We will amend our paper to better explain the experiments.
**R1.2 Message candidates proposal and tractability.** Please refer to R2.2.
**R1.3 Argmax for RSA final decision.** Actually, we used this setting in the experiment, but it does not outperform IBR.
**R1.4 Time scales (Lines 44-47, 84-86, 113, 138, 144).** Gricean pragmatic maxims pertain to conscious rational and cooperative actions of humans.[1] According to Lewis et al. (2014), cached equilibria from repeated interactions on the pragmatic timescale give rise to changes on the developmental timescale. "Stateless" means the target and distractors remain the same for each short-term instance, in which agents' emergent parameters ($P_{S_0}$ and $P_{L_0}$) can be seen as steady priors for (multi-round) pragmatic reasoning methods. $P_{S_k}$ is the $k$-th level reasoning strategy in RSA/IBR.
**R1.5 SampleL/ArgmaxL/IBR (Line 120, 147)** Listeners are probabilistic in SampleL and deterministic in ArgmaxL; speakers are deterministic in both; indeed, ArgmaxL is better. In fact, Andreas & Klein (2016) corresponds to SampleL. For SampleL (0.5), speakers are deterministic (=IBR S1) and listeners probabilistic (=IBR L0, before reasoning starts).
**R1.6 Typos / expressions / citations in Lines 35, 109, 168, 328 and Figure 3.** Thanks, we will correct these.
**R1.7 Reproducibility.** We submitted the code as well as documentation to enable reproducibility.

**R2.1 Natural senarios.** We also noticed Lazaridou et al. (ACL 2020) posted after the NeurIPS submission, which we will cite. However, this paper focuses on insights that vary from theirs (see R1.1). Natural linguistic tasks are clearly our future direction and we have initial results on figurative language, but these may appear in another paper.
**R2.2 Tractability.** All pragmatic methods in Table 1 use the same message candidate set (i.e., 75% sub-sampling, "non-trivial"). This usually leads to a small message set size $|M_\cup|$, so the problem is tractable. We agree that investigating scalability to large $|M_\cup|$ is important, and the incremental pragmatics of Cohn-Gordon et al. (2018, 2019) are inspiring: similar to RSA, we can also apply GameTable(-s) in each time step in an *unrolling* procedure. Within each time step, our time complexity analysis still holds, which may make GameTable(-s) more tractable. Other ways to reduce $|M_\cup|$ are also possible, e.g., hierarchically classifying messages and applying pragmatics for each decision layer.
**R2.3 AlexNets.** They are used because the visual perception tasks here are relatively easy.

**R3** Consistency with habits is important for realistic language scenarios, as humans typically prefer common words over obscure ones. Most discussions in Sec. 4.2 also hold for Sec. 4.3. We will add an appendix section on environments.

**R4** Distinct AlexNets: Transformation from the output of one to another is highly non-linear. "Hierarchy depth" refers to the iteration rounds of "$S$ thinks that $L$ thinks that $S$ thinks that $L$ thinks that...".

## Footnotes

[1] https://plato.stanford.edu/entries/pragmatics/


[Meta-Review · NeurIPS 2020]

All reviewers agree that this is an interesting, sound submission above acceptance threshold. I have read the reviews and author response and I would like to propose acceptance. Specifically, I agree with R1 that the idea of considering explicit equilibria methods in the context of multi-agent communication will inspire more research in the field. Moreover, the application on Starcraft domain is also a good contribution and overall this work provides good accuracy-based improvements with the proposed pragmatic reasoning method. However, I agree with R1 that a discussion beyond accuracy results (e.g., looking at the intrinsic properties of the learned communication) would have been really helpful. At the same time, the reviewers have raised a number of concerns, most of which appear have been clarified in the author response. In particular I think clarifying in the final manuscript the points raised by R1 and R2 (and the subsequent author responses) would improve the clarity of the paper.